# LANGUAGE MODELS THAT THINK, CHAT BETTER

## ABSTRACT

Reinforcement learning with verifiable rewards (RLVR) improves language model reasoning by using rule-based rewards in verifiable domains such as mathematics and code. However, RLVR leads to limited generalization for open-ended tasks—such as writing outline essays or making meal plans—where humans reason routinely. This paper shows that the RLVR paradigm is effective beyond verifiable domains, and introduces **RL** with **M**odel-rewarded **T**hinking (**RLMT**) for general-purpose chat capabilities. Using diverse real-world prompts, RLMT requires LMs to generate long CoT reasoning before response, and optimizes them with online RL against a preference-based reward model used in RLHF. Across 40 training runs on Llama-3.1-8B and Qwen-2.5-7B (both base and instruct) and multiple optimization algorithms (DPO, PPO, and GRPO), RLMT consistently outperforms standard RLHF pipelines. This includes substantial gains of 3–7 points on three chat benchmarks (AlpacaEval2, WildBench, and ArenaHardV2), along with 1–3 point improvements on other tasks like creative writing and general knowledge. Our best 8B model surpasses GPT-4o in chat and creative writing and rivals Claude-3.7-Sonnet (Thinking). RLMT can also be applied directly to base models without an SFT stage, akin to R1-Zero training (DeepSeek-AI, 2025). Remarkably, with only 7K prompts, Llama-3.1-8B base trained with our RLMT recipe outperforms Llama-3.1-8B-Instruct post-trained with a complex multi-staged pipeline with 25M+ examples. We close with qualitative and quantitative analyses of how trained models plan their responses. Our results rethink the post-training pipeline and call upon future work to understand and employ thinking more broadly.[1]

## 1 INTRODUCTION

Thinking through the consequences of one's actions—and revising them when needed—is a defining feature of human intelligence (often called "system 2 thinking", Kahneman (2011)). It has also become a central aspiration for large language models (LLMs). Recent progress toward this goal has been driven by reasoning models trained through reinforcement learning with verifiable rewards (RLVR; Lambert et al., 2025; DeepSeek-AI, 2025). In RLVR, models are optimized with automatically checkable rewards from domains such as mathematics and code, encouraging them to reason with a long chain-of-thought (CoT; Nye et al., 2021; Wei et al., 2022) before answering.

So far, RLVR has been applied to math, coding (DeepSeek-AI, 2025; Zeng et al., 2025), STEM problems (Ma et al., 2025), and to a lesser extent other deterministic puzzles and games (Chen et al., 2025; Liu et al., 2025b; Stojanovski et al., 2025; Liu et al., 2025c). Although humans rely on reasoning in everyday tasks such as writing emails, drafting essay outline, and making to-do lists, we find that the skills acquired from RL in verifiable domains do not naturally transfer to these general tasks. Figure 3 shows that open-source reasoning models trained via math-focused RLVR lag behind standard instruction-tuned models on WildBench (Lin et al., 2025b), a widely used chat benchmark with diverse user queries. Complementary studies report limited generalization of RLVR-trained models to reasoning tasks beyond verifiable domains (Huan et al., 2025; Zhou et al., 2025).

This paper pushes the RLVR paradigm well beyond verifiable domains to general-purpose chat, and introduces **R**einforcement **L**earning with **M**odel-rewarded **T**hinking (**RLMT**). As in Figure 1,

---

[1]We release our code and models at ⟨redacted for submission⟩. An anonymized version of the code is zipped with this submission.

Figure 1: We train LMs with long chain-of-thought on diverse, general user prompts through reinforcement learning with a reward model. RLMT allows models to think compared to RLHF, and extends RLVR to broader, open-ended tasks.

RLMT trains LMs to generate long CoT reasoning before final answers, using online RL algorithms such as GRPO (Shao et al., 2024). Unlike RLHF (Ziegler et al., 2020; Ouyang et al., 2022), this

design encourages explicit reasoning. Compared to RLVR relying on rule-based rewards tied to ground-truth answers, RLMT only requires prompts and uses reward models trained on human preference data over diverse prompts, as in RLHF, to evaluate responses. RLMT recipe is surprisingly effective across a wide range of tasks, enabling long CoT for open-ended tasks (see Figure 2 for an example).

RLMT is effective for both base models and those warm-started with a supervised fine-tuning (SFT) stage, likened to R1-zero and R1 (DeepSeek-AI, 2025). We apply this recipe to two model families: Llama-3.1-8B and Qwen-2.5-7B. We begin with models that undergo SFT on thinking traces and responses generated by Gemini 2.5 Flash (Comanici, 2025), and then optimize these warm-started models with RL against a reward model, specifically Skywork-v2 (Liu et al., 2024a). Across both families and multiple optimization algorithms (on-policy DPO, PPO, and GRPO), RLMT produces consistent and sizable improvements over standard RLHF, with average gains of 3–7 points on different chat benchmarks, and 1–3

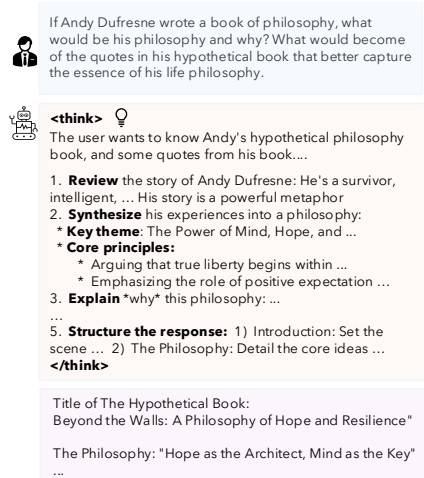

Figure 2: Example reasoning trace generated by an LM trained with RLMT for an open-ended query.

points on other tasks including creative writing and general knowledge. Our best model—Llama-3.1-8B-Instruct trained with RLMT (GRPO)—scores 58.7 on AlpacaEval2 and 50.4 on WildBench (Table 1). It comfortably surpasses models $10\times$ larger (Llama-3.1-70B-Instruct and Qwen2.5-72B-Instruct), and even beat GPT-4o (OpenAI et al., 2024) and Claude-3.7-Sonnet (Anthropic, 2025) on WildBench.

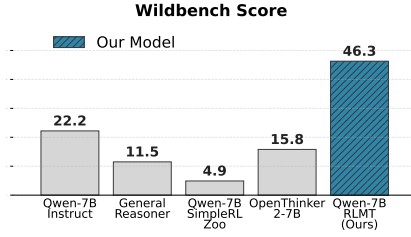

Figure 3: Score on WildBench (Lin et al., 2025b). Thinking models trained only on verifiable domains do not generalize well to general-purpose chat.

Furthermore, RLMT delivers substantial gains even when applied directly to base models without an SFT stage. In this setting, RLMT achieves average chat scores of 15.6 on Llama-3.1-8B and 29.0 on Qwen-2.5-7B (Table 1). These numbers are higher than those of Llama-3.1-8B-Instruct and Qwen-2.5-7B-Instruct by more than 5 points, despite the latter relying on far more complex post-training pipelines involving millions of examples, rejection sampling, and iterative preference optimization (Llama3, 2024).

We conclude with extensive analyses that surface several interesting findings. One is the difference in pre-RL vs. post-RL performance across model families: while Llama-3.1 underperforms Qwen-2.5 before RL, the trend reverses afterwards. We hypothesize that RLMT helps reinforce certain capabilities in models, even if they are not fully optimized during pre-training or SFT. We then quantify a shift in Llama-3.1-8B's reasoning style after RL—from linear, checklist-style outlines to richer behaviors such as constraint enumeration, theme grouping, and iterative refinement. Ablation studies reveal that the choices of both the prompt mixture and the reward model are critical to the final performance. Our results indicate sufficient promise in the long-CoT paradigm for future work to undertake more detailed analyses of the models it yields.

## 2 REINFORCEMENT LEARNING WITH MODEL-REWARDED THINKING

### 2.1 BACKGROUND

We first set up the preliminary background on two LM training paradigms, RL from Human Feedback (RLHF, Ziegler et al. (2020) and RL with Verifiable Rewards (RLVR, Lambert et al. (2025)).

**RLHF.** The goal of RLHF is to align LM outputs with human preferences. Let $\pi_\theta$ denote a language model with parameters $\theta$. Given a prompt $x \sim \mathcal{X}$, the LM generates a response $y \sim \pi(\cdot \mid x)$. Let $r$ denote a reward function that assigns a scalar score to response $y$ for prompt $x$, i.e., $r(y, x) \in \mathbb{R}$. In practice, $r$ is instantiated as a reward model trained on human preference data, so that higher scores correspond to outputs better aligned with human judgments. RLHF optimizes $\theta$ to maximize the expected reward of responses generated from $\pi_\theta$:

$$\max_\theta \mathbb{E}_{x \sim \mathcal{X}} \left[ \mathbb{E}_{y \sim \pi_\theta(\cdot \mid x)} r(x, y) \right] \tag{1}$$

**RLVR.** RLVR has become the de facto method for training LMs in domains where ground-truth verification is possible, such as mathematics or code. RLVR modifies the RLHF framework by replacing the model-based reward $r$ with a verification function; for example, the indicator function $\mathbb{1}\{y = y^*\}$ against a ground-truth answer $y^*$. In practice, the verification function may go beyond simple equality checks (e.g., using unit tests for code generation).

Another distinction of RLVR from RLHF is that the LMs usually first produce a reasoning trace $z \sim \pi_\theta(\cdot \mid x)$ before a response $y \sim \pi_\theta(\cdot \mid x, z)$ (DeepSeek-AI, 2025), instead of directly generating responses. The optimization objective then maximizes the expected correctness of the final response:

$$\max_\theta \mathbb{E}_{x \sim \mathcal{X}} \left[ \mathbb{E}_{(y,z) \sim \pi_\theta(\cdot \mid x)} \mathbb{1}\{y = y^*\} \right]. \tag{2}$$

For both RLHF and RLVR, there is a variety of RL or on-policy preference learning algorithms that can be used. In this work, we focus on three widely adopted methods (DPO, PPO, and GRPO). We provide more details in Appendix E.

### 2.2 RLMT: COMBINING RLHF AND RLVR

While recent RLVR models achieve strong results in formal domains, they exhibit limited generalization to broader reasoning problems (Huan et al., 2025; Zhou et al., 2025) and chat benchmarks (see Figure 3 and results in §4). Meanwhile, planning and reasoning do help human perform a wide range of day-to-day tasks.

We propose reinforcement learning with model-rewarded thinking (RLMT) to employ broad supervision for open-ended tasks. RLMT optimizes LMs with the following objective:

$$\max_\theta \mathbb{E}_{x \sim \mathcal{X}} \left[ \mathbb{E}_{(y,z) \sim \pi_\theta(\cdot \mid x)} r(y, x) \right]. \tag{3}$$

As in Eq (3), RLMT requires LMs to generate a reasoning trace $z$ before producing the final response $y$, which differs from RLHF, and uses a reward model $r$ to score responses, rather than rule-based verification as in RLVR. We study **several key design choices** for RLMT:

**Training algorithm.** We experiment with different RL algorithms: on-policy DPO (Rafailov et al., 2023), PPO (Schulman et al., 2017), and GRPO (Shao et al., 2024).[2] The choice of training algorithm leads to different performance outcomes (§3). Our best-performing models are trained with GRPO, but our models remain better than baselines in all settings.

**Reward model.** We adopt `Skywork-v1-Llama-3.1-8B-v0.2` (Liu et al., 2024a) as our reward model $r$, which has shown strong performance on reward benchmarks (Liu et al., 2024b) and downstream applications (Malik et al., 2025). We find that having a strong reward model is instrumental for RLMT (ablations in Section 4).

---

[2]Unlike standard DPO using a static preference dataset, we sample preference pairs using the policy model.

**Prompt mixture.**    We construct the prompt distribution from diverse, real-world user requests. Concretely, we use 7.5k prompts from the *WildChat-IF* subset of the Tülu3 SFT mixture.[3] This subset prioritizes conversational prompts sampled from WildChat (Zhao et al., 2024), covering a wide range of realistic user queries. In contrast to the full Tülu-3M SFT mixture that contains a high proportion of math and jailbreak prompts, using WildChat-IF allows us to better capture general usage. Analysis shows that this choice improves general-purpose chat performance over alternatives such as UltraChat (Cui et al., 2024); see §4 for details.

## 2.3    WARM-START SFT TRAINING AND "ZERO" TRAINING

Since the LMs we use do not naturally adopt the desired thinking format, we try two methods to elicit this behavior: (1) warm-starting with supervised fine-tuning (SFT), and (2) directly prompting base models without SFT (the "Zero" approach of DeepSeek-AI (2025)).

**Warm-start thinking with SFT.**    We begin by teaching models the desired thinking format via supervised fine-tuning (SFT). Specifically, we sample 6k prompts from the Tülu3 SFT mixture (disjoint from those prompts used for RLMT) for SFT. We generate responses using Gemini 2.5 Flash (0417 Preview), a popular teacher model in recent approaches that distill reasoning behavior from reasoning models (Muennighoff et al., 2025; Guha et al., 2025). Since Gemini's CoT is not accessible, we prompt it to produce a simulated thinking trace before the final response. We additionally experiment with SFT data generated by GPT-4.1-mini and observe similar results and trends (see §4.1). We list details of hyper-parameters in Appendix B and prompt formats in Appendix D.

**Zero training with base models.**    We also directly apply RLMT to base models without a warm start, which we refer to as the *Zero* setting. Concretely, we experiment with Llama-3.1-8B (Llama3, 2024) and Qwen-2.5-7B (Qwen-2.5, 2025), neither of which has undergone post-training. In this case, we elicit the desired output structure by prepending a fixed instruction prefix (A conversation between User and Assistant...", see Appendix D). The subsequent RL training procedure is otherwise identical to the setup described for RLMT.

# 3    THINKING BENEFITS OPEN-ENDED REASONING

## 3.1    SETTINGS, BENCHMARKS AND EVALUATED MODELS

**Setting and models.**    We evaluate RLMT in two settings: (1) applied to models after SFT warm-start (Tables 1 and 3), and (2) applied directly to base models (the "zero" setting; Tables 1 and 3). We apply setting (1) with SFT on top of the Base and Instruct versions of Llama-3.1-8B and Qwen2.5-7B. Following DeepSeek-AI (2025), we apply the zero training (2) only to base models. In Table 1, we report GRPO as our main results, since it achieves the best overall performance and serves as the basis for our analysis. We provide results with DPO and PPO in Table 3 for comparison.

**Benchmarks.**    We evaluate our models on a suite of 7 benchmarks spanning general chat, creative writing, instruction following, and general knowledge—these are chosen to represent a meaningful selection of broadly applicable tasks. We list the benchmarks below:

1. **Chat.**    We include the widely used **WildBench (WB)** (Lin et al., 2025b), **AlpacaEval 2 (AE2)** (Dubois et al., 2024), and **ArenaHardV2 (AH2)** (Li et al., 2024a;b) for chat evaluation. AE2 and AH2 uses a free-form judgment procedure, whereas the WB relies on carefully crafted rubrics.
2. **Creative writing.**    We augment these benchmarks with **CreativeWritingV3 (CWv3)** (Paech, 2025) to evaluate the creative writing abilities of our models. The WB score ranges from -100 to 100, while the other three range from 0 to 100.
3. **Instruction following.**    We use the recently published **IFBench (IF$_{Ben}$)** benchmark (Pyatkin et al., 2025) to produce a score ranging from 0–100.
4. **General knowledge.**    We evaluate our models on **MMLU-Redux (MMLU$_R$)** (Gema et al., 2025) and **PopQA** (Mallen et al., 2022) to test general and long-tail knowledge, respectively. The resulting scores span 0–100.

---

[3]https://huggingface.co/datasets/allenai/tulu-3-wildchat-if-on-policy-8b

Table 1: GRPO results for models trained from Llama-3.1-8B and Qwen2.5-7B (base and instruct) in both warm-started and zero settings. 💡 shows whether thinking was enabled, with ✓ denoting RLMT models and × denoting RLHF models. The best numbers are **bolded** in each category. Our main focus is on chat benchmarks: WildBench (WB), AlpacaEval2 (AE2), and ArenaHardV2 (AH2). When evaluating un-trained base models, we prompt them with both thinking and non-thinking template (tpl). Thinking models outperform non-thinking baselines, especially on chat and creative writing.

| Backbone | Training | 💡 | WB | AE2 | AH2 | $Avg_{Chat}$ | CWv3 | PopQA | $IF_{Ben}$ | $MMLU_R$ | Avg |
|---|---|---|---|---|---|---|---|---|---|---|---|
| *SFT Warm-Started Models* | | | | | | | | | | | |
| Llama-3.1-8B | + SFT | × | -10.6 | 26.8 | 5.1 | 7.1 | 75.2 | 26.4 | 15.6 | 59.7 | 28.3 |
| | | ✓ | -1.6 | 29.7 | 6.5 | 11.5 | 75.1 | **30.5** | 17.0 | 61.1 | 31.2 |
| | + GRPO | × | 33.2 | 46.7 | **16.3** | 32.1 | **84.2** | 24.5 | **17.7** | 61.2 | 40.5 |
| *Llama-3.1-8B-RLMT* | | ✓ | **38.1** | **52.3** | 15.9 | **35.4** | 80.9 | 30.3 | 15.6 | **61.7** | **42.1** |
| Qwen2.5-7B | + SFT | × | -0.9 | 28.8 | 8.0 | 12.0 | 61.9 | 21.1 | 19.0 | 65.2 | 29.0 |
| | | ✓ | 12.0 | 33.9 | 10.6 | 18.8 | **69.5** | 22.0 | 21.1 | 62.1 | 33.0 |
| | + GRPO | × | 28.9 | 51.0 | 13.1 | 31.0 | 60.9 | **24.4** | 19.7 | **72.8** | 38.7 |
| *Qwen2.5-7B-RLMT* | | ✓ | **31.0** | **54.0** | **19.1** | **34.7** | 65.7 | 22.8 | **21.8** | 67.3 | **40.2** |
| Llama-3.1-8B-Instruct | | × | -7.0 | 32.1 | 5.1 | 10.1 | 55.0 | **36.4** | 23.8 | 70.0 | 30.8 |
| | + SFT | × | 12.1 | 33.5 | 9.9 | 18.5 | 78.5 | 31.1 | 23.5 | 64.8 | 36.2 |
| | | ✓ | 14.3 | 34.5 | 9.5 | 19.4 | 78.7 | 32.5 | **24.8** | 70.6 | 37.8 |
| | + GRPO | × | 42.0 | 45.6 | 19.9 | 35.8 | 83.6 | 31.8 | 21.1 | **71.7** | 45.1 |
| *Llama-3.1-8B-Instruct-RLMT* | | ✓ | **50.4** | **58.7** | **22.9** | **44.0** | **84.3** | 34.0 | 22.1 | 70.0 | **48.9** |
| Qwen2.5-7B-Instruct | | × | 22.2 | 37.1 | 10.0 | 23.1 | 49.8 | 22.2 | **28.2** | **75.4** | 35.0 |
| | + SFT | × | 12.6 | 28.6 | 10.0 | 17.1 | 65.4 | 21.1 | 19.0 | 67.2 | 32.0 |
| | | ✓ | 18.7 | 33.7 | 10.9 | 21.1 | 71.0 | 21.8 | 21.8 | 60.5 | 34.1 |
| | + GRPO | × | 37.4 | 41.6 | 16.3 | 31.8 | 72.6 | 21.9 | 17.0 | 73.1 | 40.0 |
| *Qwen2.5-7B-Instruct-RLMT* | | ✓ | **46.3** | **50.5** | **20.8** | **39.2** | **75.6** | **22.5** | 20.1 | 71.5 | **43.9** |
| *Zero Training (No SFT)* | | | | | | | | | | | |
| Llama-3.1-8B | Base (nonthink tpl) | × | -87.6 | 1.7 | 0.8 | -28.4 | 30.2 | 25.2 | 17.0 | 36.6 | 3.4 |
| | Base (think tpl) | ✓ | -88.2 | 1.6 | 0.6 | -28.7 | 31.8 | 20.7 | 16.7 | 26.5 | 1.4 |
| | + GRPO | × | -4.8 | 29.8 | 4.5 | 9.8 | 47.5 | 29.5 | 16.0 | 55.2 | 25.4 |
| *Llama-3.1-8B-RLMT-Zero* | | ✓ | **7.2** | **34.0** | **5.6** | **15.6** | **49.0** | **31.2** | **18.0** | **56.2** | **28.7** |
| Qwen2.5-7B | Base (nonthink tpl) | × | -65.8 | 4.5 | 2.0 | -19.8 | 39.1 | 23.1 | 17.3 | 63.2 | 11.9 |
| | Base (think tpl) | ✓ | -68.4 | 4.4 | 1.5 | -20.8 | 36.7 | 23.0 | 17.3 | 59.9 | 10.6 |
| | + GRPO | × | 13.4 | 48.4 | 8.8 | 23.5 | 50.7 | **25.2** | 16.0 | **71.9** | 33.5 |
| *Qwen2.5-7B-RLMT-Zero* | | ✓ | **22.2** | **54.0** | **10.8** | **29.0** | **54.0** | 24.2 | **18.0** | 71.8 | **36.4** |

We provide more details on the evaluation process (e.g., judge model, length control) along with four more benchmarks, including math and logical puzzles, in Appendix C.

**Baselines and algorithms.** We pair every RLMT model with an RLHF baseline under the same training setup, differing only in the absence of thinking. To rigorously isolate the effect of integrating long CoT in post-training, we construct a matched set of *non-thinking* baselines trained with RLHF paradigm. For every thinking model in any setting, we train a corresponding non-thinking model that follows the same setting without thinking. Concretely, we still take prompt- response pairs distilled from Gemini thinking for a fair comparison, but we removed the thinking trace in this case. We evaluate our models and baselines with DPO, PPO, and GRPO (more details in Appendix B).

## 3.2 RESULTS

**Thinking models excel in chat and creative writing.** Table 1 contains results after SFT, and after GRPO for both thinking and non-thinking models. Thinking models trained with RLMT consistently outperform non-thinking counterparts by 1.5–4 points on average across all benchmarks (Tables 1 and 3). The gap over baselines is maximum on chat (WildBench and AlpacaEval2): 3–8 points on average. They are usually also better at creative writing and factuality (PopQA).

Table 2: Comparison of Llama-3.1-8B-Instruct-RLMT with strong open-source and closed models, including GPT-4o and Claude-3.7-Sonnet (a thinking model).

| Model | Avg. | WB | AE2 | AH2 | CWv3 |
|---|---|---|---|---|---|
| *Our model* | | | | | |
| L3.1-8B-I-RLMT | 54.1 | **50.4** | **58.7** | 22.9 | 84.3 |
| *Other models* | | | | | |
| L3.1-70B-Instruct | 32.1 | 16.3 | 42.0 | 10.6 | 59.4 |
| Q2.5-72B-Instruct | 45.2 | 44.4 | 50.2 | 19.9 | 66.3 |
| GPT-4o | 53.2 | 46.2 | 56.5 | 32.1 | 77.8 |
| Claude3.7-Sonnet | **58.9** | 47.8 | 58.1 | **39.3** | **90.3** |

To provide a further reference, Table 2 compares our best model, Llama-3.1-8B-Instruct-RLMT, to four strong models. Remarkably, despite being $10\times$ smaller than the two open source models, our RLMT model outperforms them both by large margins (9–22 points). It also **outperforms GPT-4o** on chat and creative writing. We also compare it to the frontier thinking model Claude-3.7-Sonnet. (February 2025, rumored to be 150B+ scale and post-trained on millions of examples). Llama-3.1-

Table 3: DPO/PPO results for warm-start and zero training. 💡 shows whether thinking is enabled, with ✓ denoting RLMT models and × denoting RLHF models. The best numbers are **bolded** and the second are underlined. Warm-start + RLMT remains effective with DPO/PPO, but they lag GRPO. The two algorithms are ineffective compared to GRPO for zero training.

| Backbone | Training | 💡 | WB | AE2 | AH2 | $Avg_{Chat}$ | CWv3 | PopQA | $IF_{Ben}$ | $MMLU_R$ | Avg |
|---|---|---|---|---|---|---|---|---|---|---|---|
| *SFT Warm-Started Models* | | | | | | | | | | | |
| Llama-3.1-8B | + SFT | × | -10.6 | 26.8 | 5.1 | 7.1 | 75.2 | 26.4 | 15.6 | 59.7 | 28.3 |
| | | ✓ | -1.6 | 29.7 | 6.5 | 11.5 | 75.1 | 30.5 | 17.0 | 61.1 | 31.2 |
| | + DPO | × | 14.4 | 34.6 | 9.0 | 19.3 | 78.3 | 28.2 | 17.0 | 60.3 | 34.5 |
| | | ✓ | 17.3 | 36.8 | 10.9 | 21.7 | 76.6 | 32.4 | 17.3 | 62.5 | 36.3 |
| | + PPO | × | **23.4** | 40.2 | **11.9** | **25.2** | 71.3 | 29.6 | 16.7 | 62.1 | 36.5 |
| | | ✓ | 21.7 | **43.3** | 10.7 | **25.2** | **81.9** | **33.0** | **18.0** | **64.3** | **39.0** |
| Qwen2.5-7B | + SFT | × | -0.9 | 28.8 | 8.0 | 12.0 | 61.9 | 21.1 | 19.0 | 65.2 | 29.0 |
| | | ✓ | 12.0 | 33.9 | 10.6 | 18.8 | 69.5 | 22.0 | **21.1** | 62.1 | 33.0 |
| | + DPO | × | 12.8 | 32.5 | 11.6 | 19.0 | 70.5 | 21.4 | 17.0 | **70.1** | 33.7 |
| | | ✓ | 24.9 | 38.1 | 14.1 | 25.7 | 74.8 | **22.6** | 21.1 | 66.4 | 37.4 |
| | + PPO | × | 29.0 | 35.5 | 14.9 | 26.5 | 69.7 | 21.9 | 18.0 | 68.6 | 36.8 |
| | | ✓ | **30.9** | **42.2** | **15.2** | **29.4** | **77.2** | 22.3 | 20.4 | 67.1 | **39.3** |
| Llama-3.1-8B-Instruct | | × | -7.0 | 32.1 | 5.1 | 10.1 | 55.0 | **36.4** | 23.8 | 70.0 | 30.8 |
| | + SFT | × | 12.1 | 33.5 | 9.9 | 18.5 | 78.5 | 31.1 | 23.5 | 64.8 | 36.2 |
| | | ✓ | 14.3 | 34.5 | 9.5 | 19.4 | 78.7 | 32.5 | 24.8 | **70.6** | 37.8 |
| | + DPO | × | 27.9 | 37.2 | 13.3 | 26.1 | 80.2 | 32.1 | 22.4 | 67.0 | 40.0 |
| | | ✓ | 29.7 | 45.1 | 15.4 | 30.1 | 81.6 | 33.4 | **25.2** | 70.5 | 43.0 |
| | + PPO | × | 43.4 | 50.9 | 17.8 | 37.4 | 83.2 | 32.4 | 21.1 | 69.5 | 45.5 |
| | | ✓ | **46.8** | **58.2** | **23.0** | **42.7** | **85.3** | 33.2 | 24.1 | 68.7 | **48.5** |
| Qwen2.5-7B-Instruct | | × | 22.2 | 37.1 | 10.0 | 23.1 | 49.8 | 22.2 | **28.2** | **75.4** | 35.0 |
| | + SFT | × | 12.6 | 28.6 | 10.0 | 17.1 | 65.4 | 21.1 | 19.0 | 67.2 | 32.0 |
| | | ✓ | 18.7 | 33.7 | 10.9 | 21.1 | 71.0 | 21.8 | 21.8 | 60.5 | 34.1 |
| | + DPO | × | 21.1 | 31.4 | 11.4 | 21.3 | 71.3 | 21.4 | 16.7 | 68.5 | 34.5 |
| | | ✓ | 28.1 | 37.6 | 14.9 | 26.9 | 73.7 | 22.0 | 22.8 | 66.0 | 37.9 |
| | + PPO | × | 33.1 | 37.4 | 15.3 | 28.6 | 71.1 | 22.1 | 19.0 | 71.5 | 38.5 |
| | | ✓ | **39.8** | **45.3** | **17.7** | **34.3** | **76.5** | **22.4** | 22.4 | 71.4 | **42.2** |
| *Zero Training (No SFT)* | | | | | | | | | | | |
| Llama-3.1-8B | Base (nonthink tpl) | × | -87.6 | 1.7 | 0.8 | -28.4 | 30.2 | 25.2 | 17.0 | 36.6 | 3.4 |
| | Base (think tpl) | ✓ | -88.2 | 1.6 | 0.6 | -28.7 | 31.8 | 20.7 | 16.7 | 26.5 | 1.4 |
| | + DPO | × | -74.6 | 2.4 | 0.3 | -24.0 | 26.1 | 32.5 | 13.3 | **47.3** | 6.8 |
| | | ✓ | -68.9 | 4.5 | 1.0 | -21.1 | 30.1 | **35.3** | 13.6 | 14.5 | 4.3 |
| | + PPO | × | -1.2 | 19.5 | 3.5 | 7.3 | 52.9 | 28.0 | 15.3 | 44.4 | 23.2 |
| | | ✓ | **10.4** | **26.1** | **3.8** | **13.4** | **56.6** | 31.9 | **18.7** | 31.5 | **25.6** |
| Qwen2.5-7B | Base (nonthink tpl) | × | -65.8 | 4.5 | 2.0 | -19.8 | 39.1 | 23.1 | **17.3** | 63.2 | 11.9 |
| | Base (think tpl) | ✓ | -68.4 | 4.4 | 1.5 | -20.8 | 36.7 | 23.0 | **17.3** | 59.9 | 10.6 |
| | + DPO | × | -24.5 | 21.7 | **5.4** | 0.9 | 38.3 | 25.0 | 16.0 | **69.3** | 21.6 |
| | | ✓ | **-18.3** | **24.6** | 4.9 | **3.7** | **40.4** | **25.3** | 14.6 | 68.8 | **22.9** |
| | + PPO | × | -70.0 | 2.9 | 1.2 | -22.0 | 36.9 | 21.9 | 16.3 | 49.2 | 8.3 |
| | | ✓ | -65.8 | 5.9 | 1.8 | -19.4 | 35.1 | 22.9 | 16.0 | 54.1 | 10.0 |

8B-Instruct-RLMT **rivals Claude-3.7-Sonnet (thinking) on AlpacaEval2 and Wildbench**, though it is worse on ArenaHardv2, likely due to the high proportion of math and coding.

**GRPO achieves the best performance, but RLMT remains effective with DPO and PPO.** Comparing Tables 1 and 3 certify that GRPO usually weighs in at about 1–3 points better than PPO, and about 5pp better than DPO on average. RLMT models retain (and indeed, extend) their edge over non-thinking baselines even with DPO and PPO. Not only are they better at Chat and Creative Writing, but they also show relative gains on instruction following ($IF_{Ben}$), factuality (PopQA) and world knowledge ($MMLU_R$). While preference optimization has often turned to DPO and PPO in recent years, we invite researchers to explore the trade-offs of using GRPO more in future work.

**RLMT can elicit chat capabilities even without SFT.** The zero training sections of Tables 1 and 3 summarize the results in the setting that directly applies RLMT to base models. Table 3 finds that the base models do not perform well, and do not show much improvement with DPO or PPO (esp. on Qwen2.5-7B). With GRPO (Table 1), however, the models show large improvements on all benchmarks. RLMT delivers a further 3-point improvement on average, allowing Qwen-RLMT-Zero to outperform Qwen2.5-7B-Instruct. Llama-RLMT-Zero performs just under Llama-3.1-8B-Instruct, but outperforms it on chat benchmarks by 5.5 points.

**RLMT models outperform thinking models learned from math data.** Figure 3 compares our RLMT models against "thinking" models trained on math, or distilled from DeepSeek-R1 on math prompts (more results in Appendix C). All RLMT models handily outperform them by 10–25 points on chat and creative writing. The gap remains large even after RL on the math models (Table 9).

Table 4: Ablation of the GRPO prompt mixture, SFT data source, and reward model. Rows marked with "↪" denote ablations. Best numbers are **bold**. We highlight the main model in light green.

| Ablation | Objective | ♀ | WB | AE2 | AH2 | Avg_Chat | CWv3 | PopQA | IF_Ben | MMLU_R | Avg |
|---|---|---|---|---|---|---|---|---|---|---|---|
| *Ablation: using different RL prompt mixture (GRPO prompts)* | | | | | | | | | | | |
| Prompts: WildchatIF | + GRPO | ✓ | **38.1** | **52.3** | **15.9** | **35.4** | 80.9 | 30.3 | 15.6 | **61.7** | **42.1** |
| ↪ Prompts: UltraFeedback | + GRPO | ✓ | 35.5 | 47.3 | 13.8 | 32.2 | 78.0 | **33.3** | 16.0 | 50.7 | 39.2 |
| ↪ Prompts: Tülu3 Random | + GRPO | ✓ | 22.2 | 42.7 | 11.5 | 25.5 | 76.4 | 31.0 | **16.3** | 59.5 | 37.1 |
| *Ablation: SFT warm-start with different data source (**GPT-4.1-mini**)* | | | | | | | | | | | |
| ↪ Warm-start: GPT-4.1-mini | + SFT | ✗ | -5.9 | 25.1 | 5.3 | 8.2 | 67.7 | 31.3 | 18.0 | 59.2 | 28.7 |
| ↪ Warm-start: GPT-4.1-mini | | ✓ | 7.0 | 26.6 | 7.3 | 13.6 | 72.9 | **33.1** | **19.4** | **68.2** | 33.5 |
| ↪ Warm-start: GPT-4.1-mini | + GRPO | ✗ | 38.6 | 44.8 | 16.2 | 33.2 | 82.1 | 28.2 | 16.3 | 60.5 | 41.0 |
| ↪ Warm-start: GPT-4.1-mini | | ✓ | **40.6** | **51.9** | **17.5** | **36.7** | 82.8 | 31.0 | 16.7 | 56.3 | **42.4** |
| *Ablation: Reward model (**SkyworkV2 & ArmoRM**)* | | | | | | | | | | | |
| ↪ RM: SkyworkV2 | + GRPO | ✗ | 32.6 | 47.8 | 18.8 | 33.1 | **82.2** | 32.6 | 19.4 | 55.6 | 41.0 |
| ↪ RM: SkyworkV2 | | ✓ | **40.9** | **53.2** | **22.8** | **39.0** | 80.4 | **33.7** | **19.7** | 47.7 | **42.6** |
| ↪ RM: ArmoRM | + GRPO | ✗ | -10.2 | 30.9 | 5.8 | 8.8 | 71.2 | 35.4 | 15.0 | **58.9** | 29.6 |
| ↪ RM: ArmoRM | | ✓ | -5.9 | 43.3 | 6.9 | 14.8 | 63.3 | 32.6 | 16.7 | 54.6 | 30.2 |

**Shifts in relative performance across model families.** An interesting observation is that while Llama-3.1 initially lags behind Qwen-2.5 before going through RL, it surpasses Qwen-2.5 after applying RLMT. We hypothesize that this arises because Qwen-2.5 undergoes more extensive tuning targeted at these benchmarks, whereas RLMT is able to reinforce and unlock these capabilities in models like Llama-3.1 that are less optimized initially.

# 4 ANALYSIS

In this section, we undertake several analyses and ablation studies to gain a better understanding what affects the performance of the thinking models, and in what aspects precisely they improve.

## 4.1 ABLATIONS

**Impact of prompt mixture.** We vary the prompt mixture used for GRPO to study its impact on downstream performance. To avoid confounders due to instruction tuning or due to oddities of the Qwen models (see Shao et al. (2025)), we perform this study on the Llama-3.1-8B. We experiment with two other data sources in Table 4: UltraFeedback (Cui et al., 2024), popular for preference optimization, and a randomly sampled subset of the Tulu3 SFT mixture (Lambert et al., 2025).

We see that the Wildchat-IF subset of Tülu outperforms both of these sources: we attribute this to the relatively simple prompts in UltraFeedback and the abundance of math and jailbreak (non-chat) prompts in the unfiltered Tulu SFT mixture. In effect, the prompts used for RL matter: more "chatty" and harder prompts lead to more improvements.

**Impact of warm-start data.** Since Gemini 2.5 Flash is a "thinking" model, one might wonder if our takeaways are an artifact of the specific choice of this model. To this end, we repeat a subset of our experiments (Llama-3.1-8B → SFT → GRPO) using GPT-4.1-mini to generate the warm-start data, leaving the prompts and other hyperparameters unchanged. As Table 4 shows, thinking models still outperform non-thinking models, especially on chat benchmarks. The final numbers achieved by the GRPO thinking models are roughly the same as the ones warm-started on Gemini 2.5 Flash.

**Impact of reward model.** To assess the importance of the reward model, we run RLMT (GRPO) with (1) Skywork-V2 (Liu et al., 2025a), a newer version of the Skywork reward model with carefully curated training data[4] (2) ArmoRM (Wang et al., 2024), another popular reward model used in alignment research (Meng et al., 2024). Both are based on Llama-3.1-8B backbones, with Skywork-V2 generally representing a stronger reward model than our default Skywork-V1, and ArmoRM representing a weaker one. Table 4 summarizes the results with and without thinking. We find: 1) **Stronger reward models lead to better performance.** The gap between the SkyworkV2 and ArmoRM results is large. A weaker model leads to drops on the non-chat benchmarks—especially with thinking. On the other hand, a strong reward model can maintain performance on non-chat benchmarks while also boosting chat performance. 2) **RLMT outperforms standard RLHF on**

---

[4]Most of our experiments were conducted before the release of Skywork-V2. We believe some of our results may also be pushed further with this version.

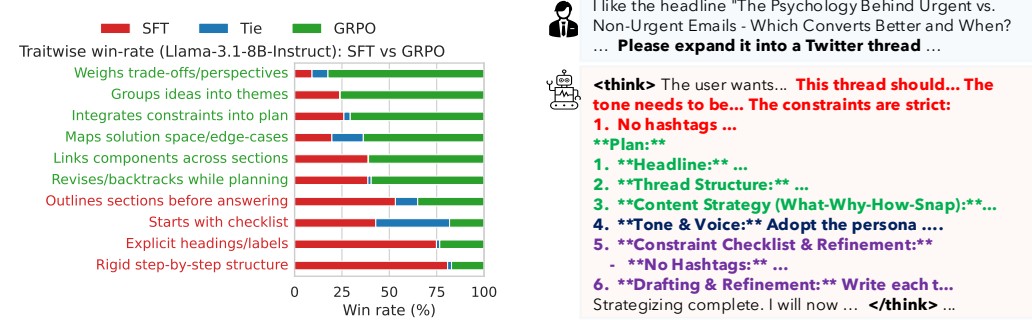

Figure 4: **Left**: Traitwise head-to-head win rates for the SFT and GRPO models. Red colors indicate that the trait diminished after GRPO, while green indicates that it increased. **Right**: Example reasoning behavior. When asked to write a tweet thread, the model first **maps out the requested constraints** and then **plans the tweet progression**. It also **runs everything through the checklist and notes necessary refinements** before producing the final output.

**chat across RMs.** With both RMs, thinking models continue to outperform their non-thinking counterparts by a large gap on chat benchmarks.

In Appendix C.3, we also compare RLMT with concurrent approaches that rely on reference-based rewards (Chang et al., 2025) or rubric-based rewards (Viswanathan et al., 2025). Our RLMT models outperform these alternatives on chat benchmarks as well.

## 4.2 How does RL training change model behavior?

**Qualitative Analysis** We analyze why precisely do the thinking models perform better on chat benchmarks. To this end, we take the best model from our suite, Llama-3.1-8B-Instruct-RLMT, and the warm-started model before RLMT. We employ the following pipeline to automatically extract the traits that maximally changed between the two versions: (1) We pass all the prompts from WildBench, and the associated thoughts generated by the two models to GPT-4.1-mini, and ask it to extract the traits most prevalent in each thought. (2) We then pass trait-sets from both models in batches of 20 prompts to GPT-4.1-mini to identify the ones consistently more prevalent in one model than the other. (3) We further summarize the identified differences across 10 random batches of 20 prompts. (4) At this point, we have a list of traits that were potentially amplified or suppressed after RLMT. For each, we compute a head-to-head win rate of which model shows the trait more across all 1024 examples of WildBench.

We plot the results in Figure 4 (left). We observe that the SFT model often starts by hierarchically planning out sections, subsections, and then use a checklist to guide the plan. On the other hand, Llama-3.1-8B-Instruct-RLMT lists out the relevant constraints and subtopics first, then groups ideas into common themes, only then does it plan out specific details. We show one (compressed) example of the thinking process in Figure 4 (right) that highlights the identified traits in the model's thoughts. We also observe that while the SFT model's planning is often linear, the RLMT model's planning is often iterative: it returns to and refines older parts of the plan, e.g., to cross-reference points mentioned elsewhere. We believe that the strategies reflected in these differences are often the traits exhibited by good writers; it is encouraging that they emerge naturally from the training process.

**Increased CoT length.** We also find that as training progresses, the model **learns to think longer** and **generate longer responses** (Figure 5), reminiscent of DeepSeek-R1-zero (DeepSeek-AI, 2025).

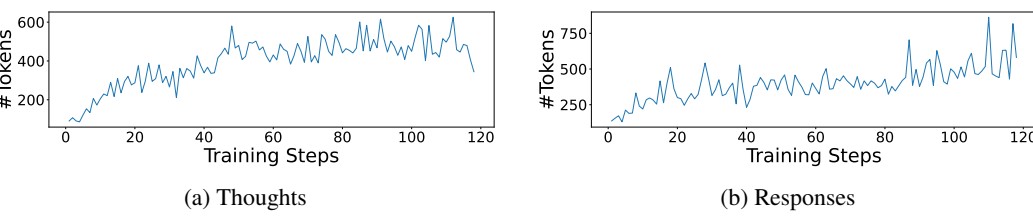

(a) Thoughts        (b) Responses

Figure 5: Llama-3.1-8B-RLMT-Zero thinks and answers longer as training progresses.

## 5 RELATED WORK

**Training stages of language models.** Modern LMs are typically trained in three stages. First, they are pre-trained on large corpora to acquire general language abilities (Vaswani et al., 2017; Devlin et al., 2019; Radford et al., 2019). Second, they undergo supervised fine-tuning (SFT) on curated prompt–response pairs (Radford et al., 2018; Taori et al., 2023; Wang et al., 2023; Llama3, 2024; Qwen-2.5, 2025), which encourages traits like instruction following (Ouyang et al., 2022; Wang et al., 2022). Finally, models are refined via reinforcement learning to enforce desirable behavior, either through preference optimization (Ouyang et al., 2022; Rafailov et al., 2023; Ethayarajh et al., 2024; Meng et al., 2024; Ahmadian et al., 2024) or domain-specific objectives, such as math (Chen et al., 2024; Kazemnejad et al., 2025) and tool use (Luo et al., 2025). In this work, we show that our simple RLMT recipe, when applied directly to base models, can also yield strongly aligned models.

**Reinforcement learning from human feedback (RLHF).** To capture subjective attributes such as human preference, RLHF relies on a learned reward model trained on pairwise human judgments (Christiano et al., 2017; Ziegler et al., 2020; Ouyang et al., 2022). Building on this foundation, preference optimization methods such as DPO (Rafailov et al., 2023), KTO (Ethayarajh et al., 2024), and SimPO (Meng et al., 2024) directly optimize models to generate preferred responses. Most approaches directly generate the response. We instead first generate internal reasoning and only later the final answer. Closely related to our work are efforts that combine preference optimization with chain-of-thought reasoning (Pang et al., 2025; Wu et al., 2025a). Unlike these approaches, which elicit long CoT traces by prompting instruct models and typically rely on offline algorithms, we show the promise of directly applying online RL with long CoT to base models.

**Reinforcement learning with verifiable rewards (RLVR).** Particularly for domains with objectively verifiable solution, more suitable algorithms have been proposed, such as GRPO(Shao et al., 2024). GRPO and its variants (Zheng et al., 2025; Yu et al., 2025) compute advantages by mean-centering rewards within a group, eliminating the need for a learned critic. Building on this, DeepSeek-R1 (DeepSeek-AI, 2025) combined GRPO with long CoT reasoning, where models generate reasoning traces that are stripped before evaluation. This paradigm enables effective test-time scaling (Snell et al., 2025), and has also been applied successfully beyond math to other reasoning domains (Liu et al., 2025c; Cheng et al., 2025; Huan et al., 2025; Ma et al., 2025). Nonetheless, RLVR remains largely confined to formal settings and has shown limited generalization to open-ended reasoning, which is the focus of our work.

**RLVR beyond rule-based rewards.** Recent efforts have extended RLVR beyond strict rule-based verification by designing alternative reward signals. In verifiable domains, reference-free signals such as entropy (Agarwal et al., 2025) or model confidence (Zhao et al., 2025) have been shown to be effective. Some other works also explored using compact models to verify responses against ground-truth answers (Ma et al., 2025; Liu et al., 2025d) or designing rewards for a specific domain (Gurung & Lapata, 2025; Wu et al., 2025b; Jia et al., 2025; Li et al., 2025). More closely related to our work, one line explores rubric-based judges (Viswanathan et al., 2025; Gunjal et al., 2025) or reference-based scores (such as BLUE) (Chang et al., 2025) for more general chat. These approaches typically do not integrate long CoT reasoning, whereas our work shows that thinking training, when paired with a strong reward model, leads to clear benefits in general chat.

## 6 CONCLUSION

We have introduced RLMT, which integrates two core components of RLVR - long chain-of-thought and online RL learning, with reward models used in RLHF. For both the base and instruct Llama-3.1-8B and Qwen-2.5-7B, across DPO, PPO, and GRPO, RLMT outperform standard RLHF by 1.5–3 points on average on a range of benchmarks spanning creative writing and general knowledge. These gains were fueled by an increase of up to 13 points on chat benchmarks, which rivaled frontier models orders of magnitude larger that were trained on millions of prompts. Surprisingly, this simple method also shows promise on the base models when skipping the SFT warm-start entirely. Finally, we analyzed the resulting models and found that (1) the RL prompt mixture plays a pivotal role in achieving good performance, and (2) desirable reasoning strategies for open-ended tasks emerge naturally from the training process. We hope that our results inspire future work to expand the horizons of current methods into more general domains.

**Reproducibility statement**  We have taken several steps to ensure the reproducibility of our results. Detailed descriptions of our models, training setup, and evaluation benchmarks are provided in Sections 2–3. Hyperparameters and prompt formats used for our experiments are included in Appendix B and Appendix C D. We release our code and models at ⟨redacted for submission⟩. **We have also included an anonymized version of our code in the zip file accompanying this submission**.

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

## A  LIMITATIONS AND FUTURE WORK

While our work finds the effectiveness of training LMs with thinking, it is unclear how much of the improvement is due to amplification of traits already present in the model, versus the learning of new traits during the SFT warm-start or RL training (Yue et al., 2025; Zhu et al., 2025).

The study of this question is important for the design of better training pipelines. It is also possible that a set of benchmarks larger than the seven considered here would lead to takeaways that were missed here—we choose a reasonable and representative set for the purpose of this paper. Since our aim was to explore how a simple method can aid model performance, we did not extensively optimize the format used for the internal CoT, the hyperparameters, or the construction of the prompt mixtures. It is possible that doing so can push our results further, and we invite future work to do so.

## B  HYPERPARAMETERS

In our warm-start experiments, we used the hyperparameters shown in Table 5 for SFT. We provide the corresponding hyperparameters for DPO and PPO/GRPO in Table 7.

We train in bf16 and enable the use of Liger-Kernel Hsu et al. (2025) for efficient training.We use the `trl` library for SFT and DPO (von Werra et al., 2020), and `verl` (`https://github.com/volcengine/verl`) for PPO and GRPO. All sampling from the actor for DPO/PPO/GRPO is done at temperature 0.7.

| Hyperparameter | Value |
| --- | --- |
| Number of datapoints | 6003 |
| Batch size | 16 |
| Num. epochs | 2 |
| Learning rate | 4e-6 |
| Warmup ratio | 0.1 |
| LR scheduler | cosine |
| Weight decay | 1e-4 |
| Adam betas | (0.9, 0.95) |

Table 5: Supervised fine-tuning (SFT) hyperparameters used in our experiments.

| Hyperparameter | Value |
| --- | --- |
| # Prompts | 7560 |
| # Responses | 8[a] |
| Reward model | Skywork-Reward-Llama-3.1-8B-v0.2 |
| Batch size | 128 |
| Num. epochs | 2 |
| Learning rate | 3e-7 |
| Warmup ratio | 0.05 |
| LR scheduler | cosine |
| Weight decay | 1e-4 |
| Adam betas | (0.9, 0.95) |
| DPO $\beta$ | 0.1 |

Table 6: Direct Preference Optimization (DPO).

[a]Sampled from the initial model before DPO.

| Hyperparameter | Value |
| --- | --- |
| # Prompts | 7560 |
| Batch size | 64 |
| Samples per prompt | 8 |
| Group size | 8[g] |
| Max prompt length | 1024 |
| Max response length | 4096 |
| Num. steps | 120 (1 epoch) |
| Actor learning rate | 1e-6[a] |
| Critic learning rate | 1e-5[p] |
| Weight decay | 0.01 |
| Scheduler | constant |
| Warmup ratio | 0 |
| Advantage estimator | GRPO/GAE |
| KL coefficient | 0.001 |

Table 7: PPO / GRPO. Entries marked [p] or [g] are only used for PPO or GRPO, respectively.

[a]We used 3e-7 for the warm-started instruct models.

## C  ADDITIONAL RESULTS

We present in this appendix results on benchmarks beyond those used in the main text.

### C.1  BENCHMARK DESCRIPTIONS AND ADDITIONAL EVALUATION

We evaluate the models trained in this paper on the following benchmarks:

1. **[WB] WildBench** (Lin et al., 2025b) evaluates models on their ability to converse with users. It has 1,024 user prompts, some of which involve multiple turns. Unlike AlpacaEval2, WildBench is evaluated using an instance-wise manually checked rubric which is less susceptible to reward hacking. Each response is compared against a reference response from GPT-4, and scored one of -100 (much worse), -50 (worse), 0 (similar), 50 (better), or 100 (much better). The final score is the mean of the instance-wise scores.

2. **[AE2] AlpacaEval2** (Dubois et al., 2024) has 805 user prompts paired with reference responses from GPT-4-1106-preview. It outputs a head-to-head win rate between 0–100% as generated by a generative judge. We use the length-controlled win-rate as recommended dby Dubois et al. (2024), but replace the default GPT-4 judge with GPT-4o.

3. **[AH2] ArenaHardV2** (Li et al., 2024a;b) has 500 challenging real-world user queries. We use the setting with a GPT-4.1 judge and style control to mitigate potential bias.

4. **[CW3] Creative Writing V3** (Paech, 2025) evaluates models on their ability to write 96 story chapters under various constraints. We generate an absolute score betwen 0–100 using GPT-4.1 as the judge.

5. **PopQA** (Mallen et al., 2022) consists of around 14k factual questions about popular and less known entities. The result is a percentage score between 0–100%.

6. **[IF$_{\text{Ben}}$] IFBench** (Pyatkin et al., 2025) provides models with 294 prompts from varied domains that each has multiple constraints such as "include three numbers in your 22-nd sentence." We average the compliance rate across all examples to produce a score between 0–100.

7. **[MMLU$_{\text{R}}$] MMLU-Redux** (Gema et al., 2025) is a manually cleaned version of the MMLU (Hendrycks et al., 2021) benchmark, consisting of 5,700 questions that test the model's ability to answer general knowledge questions across 57 subjects.

8. **IFEval**, *only in Appendix C*, (Zhou et al., 2023) provides models with 541 simple questions under a set of constraints such as "do not use commas," and generates a score between 0–100 that signifies how well the model follows instructions. This is an alternate evaluation of instruction-following capabilities.

9. **WildBench v.s.  Gemini 2.5 Flash Preview 0520 (WildBench-G,** *only in Appendix C*), (Lin et al., 2025b) is WildBench, but evaluated against reference responses from Gemini 2.5 Flash Preview 0520. Our motivation for including this is that beyond a certain number, being X% better than GPT-4-Turbo's responses (the default reference) may be less correlated with actual improvements.

10. **MATH-500,** *only in Appendix C*, (Lightman et al., 2023) consists of 500 hard math problems filtered from the MATH dataset (Hendrycks et al., 2021) by OpenAI. We report the exact match accuracy from 0–100%.

11. **ZebraLogic,** *only in Appendix C*, (Lin et al., 2025a) tests language models on 1,000 logical grid puzzles. We report the exact match accuracy from 0–100%.

We list the corresponding results in Table 8. We observe that

1. Neither the thinking nor non-thinking models perform well on IFEval. Therefore we attribute blame here to the reward model: a preference model finds it difficult to determine if specific instructions like "do not use commas" are followed. Lacking a reliable reward model, GRPO is (and other objectives are) not able to optimize for the desired behavior.

2. We see large gaps on WildBench-G, consistent with our findings in the main text. In fact, the gaps here are larger by 0.5–1 points; the difference in quality is more important when the reference is better.

3. While prior work optimizing user preferences (Rafailov et al., 2023; Meng et al., 2024) found that it tanked scores on MATH-500, we find that our post-training actually improves mathematical abilities compared to the SFT models. On the other hand, the thinking and non-thinking models perform similarly (and the latter sometimes even outperforms the former)—therefore, building a thinking post-training pipeline for all domains constitutes a useful direction for future work.

4. While the initial models perform poorly on ZebraLogic, our thinking post-training shaves a further 1–3 points off the scores.

Table 8: Additional benchmark results for all model variants.  indicates long chain-of-thought. Best numbers are **bold** and second best are underlined within each model group.

| Backbone | Objective | 💡 | IFEval | WB-G | MATH | Zebra |
|---|---|---|---|---|---|---|
| Llama-3.1-8B | + SFT | × | 50.5 | -63.3 | 13.0 | **8.4** |
| | | ✓ | **56.0** | -61.3 | 18.8 | 6.2 |
| | + DPO | × | 51.4 | -51.8 | 15.6 | 7.6 |
| | | ✓ | 45.1 | -50.5 | **21.2** | 7.7 |
| | + PPO | × | 42.0 | -51.8 | 17.6 | 7.2 |
| | | ✓ | 43.4 | -51.4 | 20.4 | 7.3 |
| | + GRPO | × | 36.8 | -34.4 | 18.8 | 5.9 |
| | | ✓ | 36.4 | **-27.1** | 19.4 | 7.0 |
| Qwen2.5-7B | + SFT | × | 46.6 | -58.8 | 60.8 | 8.7 |
| | | ✓ | 56.4 | -55.0 | 63.0 | 7.8 |
| | + DPO | × | 44.7 | -56.2 | 63.2 | **9.0** |
| | | ✓ | **57.1** | -47.5 | 64.8 | 7.7 |
| | + PPO | × | 46.0 | -45.4 | 61.6 | 8.2 |
| | | ✓ | 50.3 | -45.5 | **66.0** | 7.9 |
| | + GRPO | × | 38.4 | -46.4 | 65.0 | 7.3 |
| | | ✓ | 46.8 | **-33.3** | 64.4 | 6.7 |
| Llama-3.1-8B-Instruct | | × | **75.6** | -65.8 | 45.2 | **13.3** |
| | + SFT | × | 70.8 | -53.3 | 28.8 | 10.3 |
| | | ✓ | 72.3 | -51.4 | 40.6 | 11.0 |
| | + DPO | × | 71.2 | -44.2 | 36.2 | 11.0 |
| | | ✓ | 69.3 | -42.3 | 43.0 | 11.6 |
| | + PPO | × | 62.5 | -27.1 | 40.6 | 10.8 |
| | | ✓ | 54.9 | -18.6 | 41.2 | 10.9 |
| | + GRPO | × | 61.2 | -27.5 | **50.6** | 8.8 |
| | | ✓ | 57.1 | **-15.8** | 45.2 | 10.8 |
| Qwen2.5-7B-Instruct | | × | **71.5** | -59.2 | **74.6** | 10.3 |
| | + SFT | × | 61.6 | -56.2 | 54.0 | 7.5 |
| | | ✓ | 66.4 | -52.4 | 62.6 | 8.9 |
| | + DPO | × | 62.1 | -53.3 | 64.4 | 8.8 |
| | | ✓ | 65.1 | -46.8 | 63.0 | 9.3 |
| | + PPO | × | 63.8 | -42.7 | 64.6 | 9.2 |
| | | ✓ | 65.4 | -35.6 | 63.4 | 8.7 |
| | + GRPO | × | 63.0 | -38.9 | 65.8 | 8.5 |
| | | ✓ | 60.1 | **-31.6** | 65.0 | 7.4 |

## C.2 COMPARISON OF OUR RLMT MODELS AGAINST MODELS TRAINED ON MATH

Several models have tried replicating DeepSeek-R1's success on math and other reasoning domains via distillation, RL, or a combination of the two. Do they perform as well on chat and creative writing? We compare the four RLMT models we train with the prominent models in this space: (1) DeepSeek-R1-Distill (Llama and Qwen) (DeepSeek-AI, 2025), (2) Q2.5-7B-SimpleRL-Zoo (Qwen) (Zeng et al., 2025), (3) DS-R1-Distill-Q-7B (Qwen) (DeepSeek-AI, 2025), and (4) OpenThinker2-7B (Qwen) (Guha et al., 2025). We list these models in Table 9. We see that these models do not perform well on chat or creative writing. All RLMT models outperform all math models by margins of 10–25 points on average. While training OpenThinker2-7B—the leading math model—with RLMT yields some improvement, our models remain comfortably better.

Table 9: Comparison of RLMT models with math-trained (thinking) models.

| Model | Avg. | WB | AE2 | AH2 | CWv3 |
|---|---|---|---|---|---|
| *(Models sharing similar backbones)* | | | | | |
| **Our models** | | | | | |
| Llama-3.1-8B-RLMT | 46.8 | 38.1 | 52.3 | 15.9 | 80.9 |
| Qwen2.5-7B-RLMT | 42.4 | 31.0 | 54.0 | 19.1 | 65.7 |
| Llama3.1-8B-Inst-RLMT | **54.1** | **50.4** | **58.7** | 22.9 | **84.3** |
| Qwen2.5-7B-Inst-RLMT | 48.3 | 46.3 | 50.5 | 20.8 | 75.6 |
| ***Thinking models trained on math*** | | | | | |
| DS-R1-Distill-L-8B | 19.9 | -10.2 | 23.8 | 6.1 | 60.0 |
| Q2.5-7B-SimpleRL-Zoo | 25.5 | 4.9 | 35.4 | 8.5 | 53.2 |
| DS-R1-Distill-Q-7B | 8.2 | -29.8 | 16.2 | 6.3 | 40.1 |
| OpenThinker2-7B | 37.4 | 15.8 | 47.7 | 17.5 | 68.4 |
| ↪ + RLMT | 40.5 | 21.7 | 52.7 | 18.7 | 68.8 |

Table 10: Comparison of RLMT models with concurrent work using different reward signals for RLVR

| Model | Backbone | Avg. | WB | AE2 | AH2 | CWv3 |
|---|---|---|---|---|---|---|
| *(Models sharing similar backbones)* | | | | | | |
| **Our models** | | | | | | |
| Llama-3.1-8B-RLMT-Zero | Llama-3.1-8B (Base) | 24.0 | 7.2 | 34.0 | 5.6 | 49.0 |
| Qwen2.5-7B-RLMT-Zero | Qwen-2.5-7B (Base) | 30.2 | 22.2 | 54.0 | 10.8 | 54.0 |
| Llama-3.1-8B-Inst-RLMT | Llama-3.1-8B-Instruct | 54.1 | 50.4 | **58.7** | 22.9 | **84.3** |
| Qwen2.5-7B-Inst-RLMT | Qwen-2.5-7B-Instruct | 48.3 | 46.3 | 50.5 | 20.8 | 75.6 |
| ***Models trained using BLEU-based reward or checklist-based reward*** | | | | | | |
| Llama3.1-8B-BLEUBERI | Llama-3.1-8B (Base) | 11.6 | -18.5 | 17.4 | 2.4 | 45.3 |
| Qwen2.5-7B-BLEUBERI | Qwen-2.5-7B (Base) | 19.7 | -6.5 | 29.5 | 12.8 | 42.8 |
| Qwen2.5-7B-RLCF | Qwen-2.5-7B-Instruct | 32.1 | 23.6 | 40.2 | 10.4 | 54.3 |

## C.3 COMPARISON WITH CONCURRENT RLVR WORK USING ALTERNATIVE REWARDS

Our work extends RLVR to general domains by leveraging a reward model. Concurrent efforts have also sought to move beyond verifiable domains through alternative reward designs, such as BLEU-based reward to reference responses (Chang et al., 2025) and checklist-style rewards where an LM scores outputs against LM-generated rubrics (Viswanathan et al., 2025). We compare RLMT against these open-source models of these concurrent work on open-ended benchmarks (chat and creative writing). Specifically, we include: (1) Llama-3.1-8B-BLEUBERI, (2) Qwen-2.5-7B-BLEUBERI, and (3) Qwen-2.5-7B-RLCF. Table 10 summarizes these models along with their source backbones.

Across the same backbones, RLMT consistently achieves substantially stronger results. For example, Llama-3.1-8B-RLMT-Zero outperforms Llama-3.1-8B-BLEUBERI by roughly 13 points on average across open-ended benchmarks, despite the latter leveraging reference responses. These findings suggest that a strong reward model provides robust and effective signal for online RL training in general-purpose domains, surpassing these alternative reward designs.

## D PROMPTS

In this Appendix, we provide the prompts used for various aspects of the experiments.

**Prompt used to sample the warm-start data.** We used Gemini 2.5 Flash (`gemini-2.5-flash-preview-04-17`) with the following instruction appended to each user prompt:

```
FORMAT: First showcase a detailed planning phase where you plan your response
within <think>...</think> tags.  Then produce the actual response within
<response>...</response> tags. The content within the <think>...</think> tags
should *not* refer to the fact that a planning phase was prompted - they should
refer to the user prompt only.
```

**Model output formats and corresponding prompts.**   For the warm-started models, we used the following format for the thinking models' outputs:

```
<think> Some thinking here </think> Response here
```

The non-thinking models converse as usual.

For the prompted base models, we used the following format to elicit a model response with thinking:

```
A conversation between User and Assistant. Following the User's query, the
Assistant first plans a response, and then provides the response. The internal
reasoning process is enclosed within <think> </think> tags and the response
is enclosed within <response> </response> tags, i.e., in the format <think>
reasoning process here </think> <response> response here </response>.
User: <query> ...user text... </query> Assistant:
```

The corresponding prompt for the non-thinking models is:

```
A conversation between User and Assistant. The user asks a question, and the
assistant provides the user with a response. The response is enclosed within
<response> </response> tags, i.e., <response> response here </response>.
User: <query> ...user text... </query> Assistant:
```

**Prompts used for trait extraction.**   We use the following prompt for the extraction of the initial set of traits

```
You are analyzing the hidden planning part produced by a model before its final
answer. From the planning excerpt below, infer the key characteristics of how
the planning is performed. Focus on the style and intent of the planning, not
the specific content of the question. Return ONLY a compact JSON array (no
extra text), where each element is a short string naming one characteristic.
Aim for 3-8 distinct, non-redundant items.
```

We then prompt the model to compare batches of traits from the two models as follows.

```
You will compare planning styles for model A vs model B. You are given multiple
examples.  For each, you will see the user prompt and two lists: A_plan and
B_plan. Identify 1-3 concise, consistent differences describing how A's planning
differs from B's. Focus on stylistic/strategic patterns that recur across the
provided examples. Return ONLY a JSON array of short difference statements (no
extra text).
```

To run the identified (and summarized) traits through each prompt and calculate win rates, we prompt GPT-4.1-mini as thus:

```
You are given two hidden planning excerpts from two models: A and B. For each
trait, decide which planning shows the trait MORE strongly: 'A', 'B', or 'tie'.
Return ONLY a JSON object mapping trait_keys to 'A', 'B', or 'tie' (lowercase
also accepted). Output strictly a JSON object with these keys only, each value
one of 'A', 'B', or 'tie'.
```

# E  BRIEF OVERVIEW OF PREFERENCE OPTIMIZATION ALGORITHMS

For the benefit of readers, we provide a short overview of DPO, PPO, and GRPO here; interested readers should refer to the cited papers for more details.

**Setup.**  Let $\pi_\theta$ be a policy LM with parameters $\theta$ that maps a prompt $x$ to a distribution over responses $y$. We write $\pi_{\mathrm{ref}}$ for a frozen reference policy (often the SFT model), and $r(y, x)$ for a scalar reward coming from either a reward model (*preference-based*) or a verifier (*verifiable/automatic*). We denote log-probabilities by $\log \pi(\cdot \mid \cdot)$ and use a KL penalty weight $\lambda$ and a clipping parameter $\varepsilon$ where applicable.

**DPO (Direct Preference Optimization).**  (Rafailov et al., 2023) *Objective.* Given offline pairwise preference data $D \equiv \{(x, y^+, y^-)\}$ with a preferred $y^+$ and a dispreferred $y^-$, DPO optimizes a logistic preference objective that implicitly constrains the policy to stay close to $\pi_{\mathrm{ref}}$:

$$\mathcal{L}_{\mathrm{DPO}}(\theta) = -\mathbb{E}_{(x,y^+,y^-)\sim\mathcal{D}}\Big[ \log \sigma\big(\beta\big[(\log \pi_\theta(y^+ \mid x) - \log \pi_\theta(y^- \mid x))$$
$$-(\log \pi_{\mathrm{ref}}(y^+ \mid x) - \log \pi_{\mathrm{ref}}(y^- \mid x))\big]\big)\Big],$$

where $\sigma$ is the logistic function and $\beta$ is a temperature. DPO is an *offline* preference-optimization method: no on-policy rollout is required. It reframes RLHF as a calibrated logit-difference classification against the reference, avoiding a learned value function and making training simple and stable.

**PPO (Proximal Policy Optimization).**  (Schulman et al., 2017) *Objective.* At each iteration, sample responses $y \sim \pi_{\theta_{\mathrm{old}}}(\cdot \mid x)$, compute rewards $R = r(y, x)$ (often with a per-sample KL penalty to $\pi_{\mathrm{ref}}$), estimate token- or sequence-level advantages $A$, and optimize the clipped surrogate:

$$\mathcal{L}_{\mathrm{PPO}}(\theta) = \mathbb{E}\Big[ \min\big(\rho_\theta A, \; \mathrm{clip}(\rho_\theta, 1 - \varepsilon, 1 + \varepsilon)A\big)\Big] - \lambda \, \mathrm{KL}(\pi_\theta(\cdot \mid x) \,\|\, \pi_{\mathrm{ref}}(\cdot \mid x))$$

where

$$\rho_\theta = \frac{\pi_\theta(y \mid x)}{\pi_{\theta_{\mathrm{old}}}(y \mid x)}.$$

PPO is an *on-policy* algorithm: the model rolls out new samples, gets a reward (e.g., from a reward model), uses generalized advantage estimation (GAE, Schulman et al. (2016)) to form $A$, and updates with a clipping rule for stability. KL-to-reference regularization maintains alignment and prevents reward hacking.

**GRPO (Group Relative Policy Optimization).**  (Shao et al., 2024) *Objective.* For each prompt $x$, GRPO generates a *group* of $K$ candidates $\{y_i\}_{i=1}^K$ from $\pi_{\theta_{\mathrm{old}}}$, obtain rewards $r_i = r(y_i, x)$, and compute group-centered advantages $A_i = r_i - \frac{1}{K}\sum_{j=1}^K r_j$ instead of employing GAE. Update the policy with a PPO-style clipped objective using these $A_i$ (often without a learned critic), plus an optional KL penalty:

$$\mathcal{L}_{\mathrm{GRPO}}(\theta) = \mathbb{E}_x\left[ \frac{1}{K}\sum_{i=1}^K \min\big(\rho_\theta^{(i)} A_i, \; \mathrm{clip}(\rho_\theta^{(i)}, 1 - \varepsilon, 1 + \varepsilon)A_i\big)\right] - \lambda \, \mathrm{KL}(\pi_\theta(\cdot \mid x) \,\|\, \pi_{\mathrm{ref}}(\cdot \mid x)).$$

$$(4)$$

GRPO is *on-policy* but avoids a learned value function by using a per-prompt baseline (the group mean reward). It is especially convenient when rewards are naturally comparable within a prompt (e.g., verifiable correctness or a shared reward model). In practice it pairs well with *reasoning* rollouts (sample multiple candidates per prompt, keep relative scores, and update toward better-than-average ones).

