# OpenReview forum: "Language Models That Think, Chat Better"
_ICLR.cc/2026/Conference — Submitted to ICLR 2026_

### Official Review · Reviewer_s2c8 · 2025-10-31

**Soundness:** 2
**Presentation:** 2
**Contribution:** 2
**Rating:** 4
**Confidence:** 4

**Summary:**

This paper investigates the effectiveness of incorporating long chain-of-thought (CoT) reasoning in language model post-training for general-purpose chat capabilities. The authors introduce RLMT (Reinforcement Learning with Model-rewarded Thinking), which combines long CoT generation with online RL using preference-based reward models. Experiments are conducted across 40 training runs on Llama-3.1-8B and Qwen-2.5-7B using DPO, PPO, and GRPO algorithms. The results show consistent improvements of 3-7 points on chat benchmarks (AlpacaEval2, WildBench, ArenaHardV2) compared to standard RLHF pipelines. The best 8B model reportedly surpasses GPT-4o on chat and creative writing tasks.

**Strengths:**

1.  The paper conducts extensive experiments across multiple model families (Llama-3.1-8B and Qwen-2.5-7B), training algorithms (DPO, PPO, GRPO), and settings (warm-start vs. zero-shot). This provides robust emperical insights.
2. The paper addresses an important question about whether thinking/reasoning capabilities can improve performance on open-ended tasks beyond verifiable domains like mathematics and coding. The findings are useful for posttraining practioners.

**Weaknesses:**

1. The primary weakness is that the comparison conflates two factors: (1) the presence of thinking/CoT and (2) RLMT vs RLHF paradigm. The paper frames the comparison as "RLMT (with thinking) vs RLHF (without thinking)," but this is not a correct framing. In RLHF, one can still incorporate thinking by having models generate CoT traces and then extracting only the final output for the reward model to evaluate. The current setup makes it difficult to isolate whether improvements come from thinking itself or from the specific RLMT training paradigm. A more appropriate comparison would be: (a) RLHF with thinking vs RLHF without thinking, and (b) RLMT with thinking vs RLHF with thinking.
2. The paper is primarily empirical and does not have much novel research value (for example, an industry lab can quickly get such insights by sweeping across different methods). Also, for RLMT, there are many relevant work that do RL on unverifiable tasks, such as [1] [2]. These work are not discussed in the paper.

[1] Rubrics as Rewards: Reinforcement Learning Beyond Verifiable Domains, Gunjal et al., https://arxiv.org/abs/2507.17746
[2] Checklists Are Better Than Reward Models For Aligning Language Models, Viswanathan et al., https://arxiv.org/abs/2507.18624

**Questions:**

1. Have you considered trying out RLHF with thinking (where thinking traces are generated but only final outputs are evaluated by the reward model)? This would help isolate the contribution of thinking itself versus the RLMT training paradigm.

---

> ### Author Response · Authors · 2025-11-21
> **Response by authors**
>
> Thanks for your helpful comments and feedback! Please find our clarification and answers to your questions below.
>
> **Q1**: The comparison did not isolate whether improvements come from thinking. Have you considered a version of RLHF that incorporates thinking by generating CoT traces and extracting only the final output for the reward model to evaluate?
>
> **A1**: Thank you for the question. We believe there is a misunderstanding here. RLMT is precisely RLHF with thinking - we only measure rewards on the actual response, while the CoT traces are only optimized implicitly in this process. Nonetheless, we understand that a comparison to the other setting (where we reward both the thinking and response with a reward model) might be useful. We provide such a comparison below, where we see that RLMT is clearly superior to the other method (largely on chat benchmarks).
> Model                                                                                                                   | CWv3 | MMLUR | PopQA | AE2 | WB | AH | IfBen | Avg
> ---                                                                                                                     | ---   | ---   | ---   | ---   | ---   | ---   | ---   | ---
> Llama-3.1-8B + warm-start SFT + GRPO                                                                                     | 80.9  | 61.7  | 30.3  | 52.3  | 38.1  | 15.9  | 15.6  | 42.1
> Llama-3.1-8B + warm-start SFT + GRPO (reward on both thinking + response)                                                | 77.8  | 62.1  | 30.7  | 36.9  | 19.8  |  9.2  | 18.0  | 36.4
>
> **Q2**: The paper does not have much novel research value (for example, an industry lab can quickly get such insights by sweeping across different methods).
>
> **A2**: We respectfully disagree with this interpretation. Our findings provide novel and useful insights into when long-CoT training helps. We believe the question is actively important to the current frontier of language models. For instance, consider the recent release of the strong Qwen3 models. There, a reasoning RL stage is performed, which utilizes longCoT-training, after which an SFT fusion stage introduces non-reasoning questions. Importantly, the non-reasoning responses do not use longCoT.  We use this example to illustrate that many such design choices arise in practice, and our results highlight the promise of incorporating longCoT training even in non-verifiable domains. Even if industry labs could obtain answers to such questions with a lot of experiments, the prevalence of answers in the open domain benefits the broader community.
>
> **Q3**: Many relevant works do RL on unverifiable tasks, such as [1] [2]. These works are not discussed in the paper.
>
> **A3**: Thank you for bringing up these papers. We note that [1,2] are viewed as contemporaneous work per ICLR policy. Nevertheless, we do indeed cite and discuss them in our paper. This includes a discussion in the final paragraph of Section 5, and an empirical comparison with models trained with rubrics [2] and BLEU as a reward [3] in Appendix B.3 (we copy the table below). RLMT models outperform these models by a large margin.
>
> |                             | Backbone             | WB    | AE2  | AH2  | CWv3 | Avg  |
> |-----------------------------|----------------------|-------|------|------|------|------|
> | Llama3.1-8B-RLMT-Zero (Ours) | Llama-3.1-8B (Base)   |   7.2 | 34.0 |  5.6 | 49.0 | 24.0 |
> | Llama3.1-8B-BLEUBERI [3]     | Llama-3.1-8B (Base)   | -18.5 | 17.4 |  2.4 | 45.3 | 11.6 |
> | Qwen2.5-7B-RLMT (Ours)       | Qwen-2.5-7B-Instruct |  46.3 | 50.5 | 20.8 | 75.6 | 48.3 |
> | Qwen2.5-7B-RLCF [2]          | Qwen-2.5-7B-Instruct |  23.6 | 40.2 | 10.4 | 54.3 | 32.1 |
>
>
> [1] Rubrics as Rewards: Reinforcement Learning Beyond Verifiable Domains, Gunjal et al., 2025
>
> [2] Checklists Are Better Than Reward Models For Aligning Language Models, Viswanathan et al., 2025
>
> [3] BLEUBERI: BLEU is a surprisingly effective reward for instruction following, Chang et al., NeurIPS 2025

---

### Official Review · Reviewer_Vi3K · 2025-10-31

**Soundness:** 3
**Presentation:** 3
**Contribution:** 1
**Rating:** 2
**Confidence:** 4

**Summary:**

This paper introduces RLMT, a post-training paradigm for LLMs that combines CoT reasoning with model-based reward optimization, extending reinforcement learning with verifiable rewards (RLVR) into general-purpose chat and open-ended tasks. RLMT requires models to produce detailed thinking traces before responses, which are then optimized via online RL against a reward model trained on human preferences, similar to RLHF but with forced long-form CoT. Experiments involve the Llama-3.1-8B and Qwen-2.5-7B families, both base and instruction models, across DPO, PPO, and GRPO. From the reported results, RLMT substantially and consistently outperforms standard RLHF on chat, creative writing, and general knowledge benchmarks. Analysis covers model behaviors, prompt/reward model ablations, and qualitative planning traits.

**Strengths:**

- RL with unverifiable domains is a timely topic. This addresses a longstanding limitation in generalizing explicit reasoning to open-ended tasks.
- The analysis is quite insightful to read.
- The paper is carefully written and well organized.

**Weaknesses:**

- The contributions seem a bit limited. The key difference that the paper claimed is enabling RL to work on unverifiable domains. This is achieved through substituting a verifiable reward function in RLVR with a reward model. However, the key, in my opinion, becomes how to obtain a strong reward model such that any policy model can improve its chat performance when RL with the reward model.
- The conclusion is a bit too intuitive. It is not surprising that long CoT benefits chat performance.
- The long CoT in chat might bring extra computation overhead. I understand that the chat domain is an example of an unvarifiable domain, but usually the general domain chats demands fast response and low latency. Always conducting the long thinking might not actually be what people want.
- Experiments about scaling effects should be a substantial part of the paper, but are completely missing. How does the performance change when scaling up the data size, model size, and inference budgets (token length)?
- Some observations are not properly interpreted. For example, why does long CoT help creative writing? Where does the creativity come from?

**Questions:**

- The proposed method relies heavily on the scores produced by reward model. Is it robust to reward model bias or poor reward calibration? Have the authors measured or observed any reward gaming, reward hacking (length bias, verbosity), or mismatches between human preference and model reward during RLMT?
- How does the performance change when scaling up the data size, model size, and inference budgets (token length)?

---

> ### Author Response · Authors · 2025-11-21
> **Response by authors (1/2)**
>
> Thanks for your comments and feedback! Please find the answer below.
>
> **Q1**: The conclusion is too intuitive. It is not surprising that long CoT benefits chat performance.
>
> **A1**: We respectfully disagree with this interpretation. The generalization of benefits from long CoT to reasoning-adjacent domains like logical puzzles, as well as general domains, has been an active research topic in this field, as many works show long CoTs or even short CoTs may have limited generalization power to broader domains [1,2,3,4]. Additionally, consider the fact that in the post-training of recent model releases, such as Qwen3, reasoning and non-reasoning training are done separately, where longCoT is only trained for reasoning RL [5]. In this regard, our results offer useful and substantial insights, including that longCoT training can be useful beyond traditional reasoning domains. We also provide thorough ablations and analyses on what leads to the high effectiveness of this paradigm.
>
> **Q2**: The contribution is a bit limited. The key seems to be to obtain a strong reward model such that any policy model can improve its chat performance.
>
> **A2**: We agree with the reviewer that a strong model is important to good final performance. However, as shown in Table 3, introducing thinking tokens (the main focus of our paper) brings consistent improvements over their non-thinking counterparts trained with the same reward models. In fact, this holds even with a weaker reward model (ArmoRM, Table 4 of the submission).
> Reward modeling itself is an active research area, and we expect future improvements in reward models to further enhance performance. We view such advances as orthogonal to the contribution of our work.
>
> **Q3**: The long CoT in chat might bring extra computation overhead. For general domain chat, always conducting the long thinking might not actually be what people want.
>
> **A3**: It is true that LongCot utilizes extra tokens, costing both time and compute. However, we know that model providers have and are still deploying models that always think—examples include OpenAI o3, o4. With the recent GPT-5 and 5.1, requests are routed to one of several models, which represents one way in which such a cost can be alleviated. Nonetheless, our aim is mainly to show that thinking brings sizable benefits even to domains where it was previously unclear if so (chat, creative writing), and the question of efficient model inference is beyond the scope of our paper  We also note that the CoT ht for RLMT are typically not very long (hundreds of tokens), while usual math longCoTs run as long as thousands of tokens.
>
> *Continued in part 2 of the response*

---

> ### Author Response · Authors · 2025-11-21
> **Response by authors (2/2)**
>
> **Q4**: Does the performance change when scaling up the data size, model size, and inference budgets (token length)?
>
> **A4**: This is a good question, which we answer by conducting new experiments that (1) control inference budgets,  and (2) perform RLMT with different numbers of datapoints. We cannot scale model sizes due to compute limits (scaling to 12B or larger models would require 16 or more GPUs with 80 GB of memory each).
>
> *Results on controlling inference budget*. We evaluate our RLMT model (Llama-3.1-8B-Warm-start-RLMT) under different thinking-token budgets by comparing three settings: (1) forcing a maximum of 127 thinking tokens (by closing thinking with < /think> early and forcing the model to answer), (2) our main setting, which does not constrain thinking length and yields an average of ~260 thinking tokens across benchmarks, and (3) encouraging additional thinking by appending “Wait, let’s re-analyze the plan to make a better response” whenever the model produces fewer than 513 thinking tokens. As shown in the table, restricting the model to fewer thinking tokens causes substantial drops in performance. Interestingly, encouraging the model to think more yields moderate improvements on chat benchmarks.
>
>
> | Budget     |  WB  |  AE2 |  AH2  | ChatAvg | CWv3 | PopQA | IFBench | MMLURe | OverallAvg |
> |------------|:----:|:----:|:-----:|:-------:|:----:|:-----:|:-------:|:------:|:----------:|
> | < 128      | 23.5 | 44.9 |  13.1 |   27.17 | 73.3 |  30.2 |    17.6 |   56.3 |       37.0 |
> | ~260 (Main) | 38.1 | 52.3 | 15.90 |   35.43 | 80.9 |  30.3 |   15.60 |   61.7 |       42.1 |
> | > 512       | 42.7 | 52.6 |  17.2 |   37.50 | 82.7 |  30.4 |    16.3 |   60.5 |       43.2 |
>
>
> *Results on scaling data size*. We also conduct a scaling study by varying the amount of RL training data (2K, 4K, 8K) using the WildChatIF dataset (which provides up to 8K examples). The results suggest using more training data consistently improves performance, especially on chat benchmarks.
>
> | LLama-3.1-8B-Warm-Start |  WB  |  AE2 |  AH2 | ChatAvg | CWv3 | PopQA | IFBench | MMLURe | OverallAvg |
> |-------------------------|:----:|:----:|:----:|:-------:|:----:|:-----:|:-------:|:------:|:----------:|
> | Wildchat 2K             | 37.7 | 44.1 | 14.4 |    32.1 | 80.9 |  30.2 |    16.7 |   57.7 |       40.2 |
> | Wildchat 4K             | 37.4 | 48.5 | 15.5 |    33.8 | 81.2 |  29.8 |    16.3 |   61.5 |       41.5 |
> | Wildchat 8K (Main)      | 38.1 | 52.3 | 15.9 |    35.4 | 80.9 |  30.3 |    15.6 |   61.7 |       42.1 |
>
>
> **Q5**: Why does long CoT help creative writing? Where does creativity come from?
>
> **A5**: Although the prompts in CreativeWriting3 outline several constraints in writing a chapter of a book, the key modes of evaluation include how entertaining the text is, which is evaluated by prompting a judge LLM, and catching various cases of “slop” by hand-written rubrics. For instance, one of the rubrics is “Avoids gratuitous metaphor or poetic overload”. Therefore, higher scores can be achieved by improving the presentation (e.g., not throwing everything into a bullet list, or avoiding mentioning the same three attributes of every character) or content (make sure that previous plot points are relevant).  Figure 4 shows precisely which traits were learned to achieve these objectives, and Figure 5 shows an example response. We will clarify this point in future versions.
>
> **Q6**: The proposed method relies heavily on the scores produced by a reward model. Is it robust to reward model bias or poor reward calibration?
>
> **A6**: Since RLMT utilizes a reward model, it inherits some of the attributes of RLHF, such as a possible dependence on response length or other superficial features that a reward model might prefer. In this regard, we do not expect it to be better or worse than RLHF. We view RLMT as complementary to work on mitigating reward model bias, and expect those techniques (e.g., debiasing strategies) to be applicable to RLMT as well, which we leave as future work.
>
> [1] Does Math Reasoning Improve General LLM Capabilities? Understanding Transferability of LLM Reasoning, Huan et al., arXiv 2025.
>
> [2] To CoT or not to CoT? Chain-of-thought helps mainly with math and symbolic reasoning, Sprague et al., ICLR 2025
>
> [3] Mind Your Step (by Step): Chain-of-Thought can Reduce Performance on Tasks where Thinking Makes Humans Worse, ICML 2025
>
> [4] BLEUBERI: BLEU is a surprisingly effective reward for instruction following, Chang et al., NeurIPS 2025
>
> [5] Qwen3 Technical Report, Qwen3 Team, arXiv 2025.

---

### Official Review · Reviewer_R18a · 2025-11-01

**Soundness:** 2
**Presentation:** 4
**Contribution:** 3
**Rating:** 4
**Confidence:** 4

**Summary:**

This paper introduces RLMT: performing RL on an LLM using a learned reward model, with thinking. In experiments, they perform RLMT with a modest set of prompts and compare this to baselines that simply use STF or RL without thinking as well as fixed open and closed models. They demonstrate good performance on a range of benchmarks (especially chat benchmarks) relative to both baselines and closed models. They perform a considerable set of ablations, showing good performance across choices of RL algorithm but demonstrating the importance of a strong reward model. They provide a qualitative analysis of the differences due to RLMT as well as analysis of CoT length.

**Strengths:**

Originality. To my knowledge, despite this domain receiving a great deal of attention, this precise technique, while simple, hasn’t been showcased before, and it makes good use of some newer benchmarks useful for measuring performance on chat and creative writing.

Quality. The experimentation — both the headline comparisons as well as the ablations — are fairly extensive. The headline comparisons are sensible, and the ablations make some helpful disambiguations about what’s working here.

Clarity. This paper is very well-written very understandable.

Significance. This domain is of great interest to many and has received quite a lot of attention. The main idea here is very natural and worth having results on, and the secondary results (e.g. the No SFT runs) are a natural extension of DeepSeek results and are very much worth highlighting.

**Weaknesses:**

I’m torn on how to think about the originality here. It’s a very simple extension of pretty well-understood ideas. It’s quite close to Wu et al. 2025a (as you cite), with the adoption of some more recently adopted techniques in online RL ported to LLMs as well as updated benchmark. It’d be helpful to tease apart precisely what makes this new relative to work like that in methods.

The results feel a bit overstated in parts. In particular, looking at the warm-started models (RLMT seems to work more decisively in the no sft setting), results look strongest on the chat tasks, which are most closely aligned with the prompts used during training. The paper claims strong performance on creative writing, but that only really seems to hold for instruct models. On other benchmarks, RLMT is pretty clearly doing worse than baselines. I’d suggest making this tradeoff clearer in results.

**Questions:**

Already articulated in weaknesses.

---

> ### Author Response · Authors · 2025-11-21
> **Response by authors**
>
> Thanks for your valuable comments and feedback! Please find the answer below.
>
> **Q1**:  The approach is a simple extension of well-understood ideas. It’s close to TPO (Wu et al. 2025a). It would be helpful to provide more discussion of the differences compared to related work on improving LLMs with thinking.
>
> **A1**: Thank you for the question. We would like to respectfully disagree, as our work brings both methodological improvements as well as practical insights about how to utilize long CoT for broad domains. We included a brief comparison in the related work section. Here, we expand on it with additional details and experimental results (please see the table below for an empirical comparison).  On top of TPO (Wu et al. 2025a), which prompts *instruction-tuned LLMs* and uses iterative DPO (a semi-online algorithm) to improve them, our contributions include
> (1) We use online RL (GRPO and PPO), which shows clear improvements over algorithms like DPO (as suggested in Table 1 and Table 2 in the paper). We further compare against iterative DPO (i.e., 4 iterations of 2 epochs in total, where we re-sample responses from the model after each iteration), which we use as a stand-in for TPO since the TPO implementation is not public. As shown in the Table below, RLMT (w/ GRPO) can handily outperform TPO.
> (2) Our approach can also be applied directly to base models without instruction tuning. In this setting as well, note the large gap between RLMT and TPO  in the Table below.
> (3) We also provide extensive ablations of the choice of algorithm, prompt source, and cover evaluation on more domains beyond chat (Section 4 and Appendix C of the submission). We believe these studies hold value in directing future work on simplifying and consolidating strong post-training pipelines.
> We will add these new results to a subsequent revision.
> |                              | WB    | AE2  | AH2   | ChatAvg |
> |------------------------------|-------|------|-------|---------|
> | **LLama-3.1-8B-Warm-Start** |       |      |       |         |
> | RLMT                         |  38.1 | 52.3 | 15.9 |   35.4 |
> | TPO                          |  23.7 | 43.5 |  13.9 |   27.0 |
> | **LLama-3.1-8B-Zero**        |       |      |       |         |
> | RLMT                         |   7.2 |   34.0 |  5.6 |   15.6 |
> | TPO                          | -31.2 | 15.2 |  2.2 |   -4.6 |
>
> **Q2**: Results look strongest on chat benchmarks, which are aligned with the prompts used for training. RLMT is doing worse than the baseline on other tasks.
>
> **A2**: Thank you for bringing up this point. We agree that the big chunk of improvements lies in chat benchmarks (and on creative writing for some non-instruct settings). We will acknowledge this more centrally in the next version of our paper. However, our goal is to avoid performance degradation on PopQA and MMLU; they are knowledge-intensive tasks, and RL is generally less likely to teach new knowledge v/s SFT. Similarly, unlike several previous preference optimization methods (e.g., see [1,2,3]) that see *performance drops* on math/knowledge benchmarks (e.g., see Appendix C of [2] - SimPO shows a 15% performance reduction on GSM-8K with Llama-3-8B), RLMT preserves performance pretty well, even seeing gains of 0.6% and 2.5% on MATH-500 and PopQA, respectively with Llama-3.1-8B (Tables 3 and 8 of the submission).
>
> [1] Think LLMs: General Instruction Following with Thought Generation, Wu et al., arXiv 2024
>
> [2] SimPO: Simple Preference Optimization with a Reference-Free Reward, Meng et al., NeurIPS 2024
>
> [3] Starling-7B: Improving Helpfulness and Harmlessness with RLAIF, Zhu et al., COLM 2024

---

### Official Review · Reviewer_jcCK · 2025-11-03

**Soundness:** 3
**Presentation:** 4
**Contribution:** 3
**Rating:** 8
**Confidence:** 4

**Summary:**

This paper introduces Reinforcement Learning with Model-rewarded Thinking (RLMT) which trains LLMs to generate long CoT reasoning before final answers, using online RL algorithms such as GRPO. Compared to RLVR relying on rule-based rewards tied to ground-truth answers, RLMT only requires prompts and uses reward models trained on human preference data over diverse prompts, as in RLHF, to evaluate responses. Previously RLVR is limited to structured domains like math and code and RLMT extends RL to open-ended reasoning tasks like open-ended chat.

**Strengths:**

1. The paper introduces Reinforcement Learning with Model-rewarded Thinking (RLMT)  - a new way to incorporate reasoning in LLMs. It focuses on domains other than math, code and science and focuses on reasoning needed for creative writing and chat.
2. The paper has conducted comprehensive experiments across different RL algorithms like GRPO, DPO and PPO.
3. The paper goes the additional mile of qualitative analysis of model behavior under SFT and RLMT, as well as ablations with various SFT data sources and reward models.

**Weaknesses:**

1. It would be interesting to see what happens when RLMT training is mixed with RLHF/RLVR. Would you get the best generalized model which does well on all tasks - math, science and code as well as creative writing etc?

**Questions:**

See above.

---

> ### Author Response · Authors · 2025-11-21
> **Response by authors**
>
> Thanks for your thoughtful review. Please find the answer below.
>
> **Q**: What happens when RLMT training is mixed with RLVR? Would it lead to models that do well on broader tasks (math, science, creative writing, etc)?
>
> **A**: Thank you for bringing up this point. This is a great question. In Appendix C.2, we provide preliminary results that touch on how to build models that perform well across both verifiable and non-verifiable domains. Specifically, we apply RLMT on top of OpenThinker2-7B, a model already strong on verifiable tasks (via distillation from DeepSeek-R1) and exhibiting RLVR-style thinking patterns. We observe that adding RLMT on top of OpenThinker2-7B improves performance on chat-oriented, non-verifiable benchmarks, but still falls short of the best RLMT models trained directly for open-ended reasoning. The resulting model is strong on verifiable tasks and good (though not as strong as standalone RLMT) on non-verifiable tasks.
>
> These findings suggest that simply applying RLVR followed by RLMT in sequence is not sufficient to obtain a model that excels across all domains, and point to the need for training pipelines that combine or balance multiple reward sources more effectively.  We consider the design of such a training pipeline slightly beyond the scope of the current paper, but view it as an exciting avenue for future work. Thank you for the insightful suggestion!

---

### Author Response · Authors · 2025-11-21
**Summary of new results**

We thank the reviewers for their constructive feedback. We have responded to each reviewer’s comments individually, and we summarize the main clarifications and additional experiments that further strengthen the contributions and claims of our paper below.

1. **Experiments on scaling data size and controlling inference budgets (Reviewer Vi3K).**
We added new experiments varying both the amount of RLMT training data and the inference-time thinking budget (see response to Reviewer Vi3K). The scaling study shows that RLMT’s improvements increase as training data grows. In the inference-budget study, we find that allocating more thinking tokens yields further gains on chat benchmarks.

2. **Expanded comparisons with existing methods (Reviewer R18a, S2c8).**
 We clarified and expanded our discussion comparing RLMT with recent efforts on enhancing LLMs with thinking for general instruction following. This includes a comparison with TPO, which we now provide (response to Reviewer R18a). In addition, Appendix B.3 of the original submission contains comparisons to contemporaneous methods on non-verifiable tasks like checklist-based rewards, BLEU-based rewards (see response to Reviewer S2c8). These results show that RLMT outperforms these alternatives. In the revision, we will highlight these comparisons more clearly.

---

### Comment · Area_Chair_D6eX · 2025-11-24
**Rebuttal Received - Next Steps**

Dear Reviewers,

The authors have submitted their rebuttal. Please review their responses and provide any follow-up, such as additional questions or revisions to your review.

As the initial scores varied significantly (8 to 2), I kindly encourage you to carefully consider the other reviewers' perspectives as you formulate your final recommendation.

Thank you for your contributions to this process.

Sincerely, Your AC

---

### Author Response · Authors · 2025-12-03
**Summary of the Discussion**

We thank the reviewers and area chairs for their efforts during the response period. Although reviewers did not have an opportunity to respond due to the special situation, we believe our clarifications and additional experiments sufficiently address their concerns. We provide a summary of the response below:

**Reviewer jcCK:** The reviewer suggested exploring mixed RLMT/RLVR training. We provided preliminary results (included in the appendices of our submission) and discussed insights that point toward future work on multi-reward training.

**Reviewer R18a:** The reviewer mainly suggested adding more discussions on differences from related work. We expanded our discussion and added new experiments demonstrating substantial performance gaps between our method and recent approaches (e.g., TPO), further highlighting our contributions.

**Reviewer Vi3K:** The reviewer asked questions about the significance of our findings. We clarified these points and anchored them in recent literature to illustrate the novelty of our results. We also pointed the reviewer to our ablation studies, which show that the main effectiveness comes from how we incorporate thinking, and that this benefit is orthogonal to reward-model strength.
As suggested, we additionally added scaling experiments, which show that the effectiveness of our method increases as both training data and inference budgets scale.

**Reviewer s2c8:** The reviewer’s questions are on the experimental setup and comparisons with concurrent work. We believe these come from some misunderstandings. We clarified the setup and pointed the reviewer to additional comparisons in the appendix, which demonstrate the effectiveness of our work over concurrent work.

We appreciate all the reviewers’ feedback and will integrate these clarifications and results to further strengthen the paper.

---

### Meta-Review · Area_Chair_ZsmW · 2026-01-08

**Summary:**

A significant concern is the novelty of the work, noting it as an extension of well-understood ideas or similar to existing methods like TPO. Another major concern is that the experiments mixed together the effects of thinking with those of the RLMT training setup, making it hard to tell where the improvements were really coming from. The takeaway is not particularly surprising, like longer chains of thought help chat performance. Although some of these points were partially addressed, for example by adding scaling experiments and clarifying how the work compares to related efforts, reviewers were still actively debating the paper’s overall contribution and originality.

**Reviewer Concerns:**

The authors made a good effort to address several reviewer concerns by adding scaling experiments, showing that RLMT benefits from more training data and larger inference-time thinking budgets, and by directly comparing against concurrent methods like TPO, where RLMT clearly performed better on chat benchmarks. They also ran some ablations to clarify the role of thinking showing that rewarding only the final answer is more effective than explicitly rewarding intermediate reasoning, and also clarified that performance on knowledge-heavy benchmarks is largely preserved, rather than degraded. However, some core concerns remained unresolved: at least one reviewer continued to question the paper’s fundamental originality and research value, viewing the main idea as intuitive despite the strong empirical results. Also, inference cost and latency from long-form thinking were concerns acknowledged but not fully addressed. While early experiments on mixing domains were promising, the authors conceded that a simple training pipeline still struggles to deliver strong performance across all task types, leaving balanced multi-domain reasoning as an open problem.

**Reviewer Scores:**

Reviewer s2c8 joined the discussion and also brought up the common concern of three reviewers on the novelty side. There was no score raising.

---

### Decision · Program_Chairs · 2026-01-26

Reject